# Safe handovers for every patient: an interrupted time series analysis to test the effect of a structured discharge bundle in Dutch hospitals

Rosanne van Seben,[1] Suzanne E Geerlings,[2] Jolanda M Maaskant,[3] Bianca M Buurman,[4] TIP study group

For numbered affiliations see end of article.

**Correspondence to**
Dr Rosanne van Seben;
r.vanseben@amsterdamumc.nl

## ABSTRACT

**Objective** Patient handovers are often delayed, patients are hardly involved in their discharge process and hospital-wide standardised discharge procedures are lacking. The aim of this study was to implement a structured discharge bundle and to test the effect on timeliness of medical and nursing handovers, length of hospital stay (LOS) and unplanned readmissions.

**Design** Interrupted time series with six preintervention and six postintervention data collection points (September 2015 to June 2017).

**Setting** Internal medicine and surgical wards

**Participants** Patients (≥18 years) admitted for more than 48 hours to surgical or internal medicine wards.

**Intervention** The Transfer Intervention Procedure (TIP), containing four elements: planning the discharge date within 48 hours postadmission; arrangements for postdischarge care; preparing handovers and personalised patient discharge letter; and a discharge conversation 12–24 hours before discharge.

**Outcome measures** The number of medical and nursing handovers sent within 24 hours. Secondary outcomes were median time between discharge and medical handovers, LOS and unplanned readmissions.

**Results** Preintervention 1039 and postintervention 1052 patient records were reviewed. No significant change was observed in the number of medical and nursing handovers sent within 24 hours. The median (IQR) time between discharge and medical handovers decreased from 6.15 (0.96–15.96) to 4.08 (0.33–13.67) days, but no significant difference was found. No intervention effect was observed for LOS and readmission. In subgroup analyses, a reduction of 5.6 days in the median time between discharge and medical handovers was observed in hospitals with high protocol adherence and much attention for implementation.

**Conclusion** Implementation of a structured discharge bundle did not lead to improved timeliness of patient handovers. However, large interhospital variation was observed and an intervention effect on the median time between discharge and medical handovers was seen in hospitals with high protocol adherence. Future interventions should continue to create awareness of the importance of timely handovers.

**Trial registration number** NTR5951; Results.

## Strengths and limitations of this study

► The study design, that is, interrupted time series (ITS) analysis, provides a strong quasi-experimental design to evaluate the impact of an intervention aimed at quality improvement.
► The study design, that is, ITS analysis, provided valuable information on preintervention trends, which strengthens the results.
► Sensitivity analysis provided important insight into the interhospital variation and differences in intervention effects among hospitals.
► Only the date of sending patient handovers was recorded. Knowing whether the next care provider received information would have been informative.
► It was not possible to evaluate percentages of compliance with the study protocol and the process evaluation with the project leaders might have been an overestimation.

## INTRODUCTION

As hospital stays have become shorter and full recovery often takes place at home,[1] a safe transition from hospital to home or nursing home has become more and more important. Besides, a rising number of older chronically ill patients, who move within the health-care system, requires continuity of care.[2 3] However, transitions from hospital to primary care settings are still considered a high-risk process. Patients are discharged with little coordination or follow-up and are hardly involved in their own discharge process.[4 5]

Inadequate transitions may have serious implications for patient safety and quality of care. Postdischarge adverse events, such as medication errors, can be the consequence of insufficient or lacking communication between hospital and primary care providers, thereby contributing to higher resource use and unplanned readmission rates.[6–11] In fact, unplanned readmission rates in the first month

postdischarge are as high as 20%[12] and a recent study shows that half of them are deemed preventable.[11]

The root of a safe transition from hospital to home or nursing home is a timely transfer of the medical handover, that is, a letter containing accurate medical discharge information for the next care provider.[8 13] The general practitioner (GP) can only take over responsibility for a patient safety, when receiving a medical handover containing accurate information on, for example, medications and follow-up.[13] Nonetheless, a review of Kripalani et al showed that medical handovers are often not available, lack important information or are not sent in a timely manner.[8] Also, a more recent study performed in 20 Dutch hospitals showed that in 10% of cases medical handovers were missing and the remainder was on average sent after 1 week,[14] even though unplanned readmissions most frequently occur within the first week postdischarge.[15]

Previous studies that aimed to improve patient handovers mainly focused on specific high-risk populations and targeted patient-related factors.[16–18] Although such interventions on individualised discharge planning or transitional care have been effective in reducing readmission[16 17] and postdischarge mortality rates,[18–20] organisational factors that form the basis of a safe handover should also be optimally arranged.[13 21] In fact, in order to ensure patient safety and continuity of care, early discharge planning, a structured discharge process and timely handovers might be essential.[13 21 22] Besides, given that patients are often unprepared at the time of discharge and uncertainties about aspects, such as treatment or medication, may exist,[5] patient education, for example, in terms of a proper discharge conversation, should also be an important aspect of the discharge process.[6 7]

The aim of this study was, therefore, to implement a structured discharge process, the Transfer Intervention Procedure (TIP), in eight hospitals. The TIP contains four elements: planning the discharge date within 48 hours after admission; arrangements for required postdischarge care; preparing medical, medication and nursing handovers, and a personalised discharge letter for the patient (PPDL) within 48 hours after admission; and holding a discharge conversation 12–24 hours before discharge. We tested whether the TIP improved timeliness of medical and nursing handovers and investigated the effect of the TIP procedure on length of hospital stay (LOS) and unplanned readmissions within 30 days postdischarge.

## METHODS
### Study design and setting
We evaluated the implementation of the TIP discharge bundle in an interrupted time series (ITS), which is the strongest design when a randomised controlled trial is not feasible.[23 24] The trial protocol[25] was based on the recommendations for ITS studies,[23] and we adhered to the Standards for QUality Improvement Reporting Excellence (SQUIRE) guidelines for quality improvement reporting.[26] The current study was part of a large

national programme, initiated by the Dutch Ministry of Health, Welfare and Sport (abbreviated in Dutch: VWS): 'Addressing Waste in Health Care'. This programme was set up in order to reduce inefficiencies in the provision of healthcare. As part of this programme, a TIP study group was established, comprising a study coordinator, two supervisors, one clinical epidemiologist, a policy officer from the Ministry of VWS and local project leaders from the eight participating hospitals (one university and seven regional teaching throughout the Netherlands) that implemented the TIP bundle at one of their surgical and one of their internal medicine wards.

Within an ITS, repeated observations are collected over time and divided into two segments, one before and one after implementation. Therefore, at six preintervention data collection points, measurements were conducted before implementation of the TIP and at six postintervention data collection points, measurements were conducted after implementation. During the implementation period of 2 months, no measurements were conducted. In February 2016, a kick-off meeting was held. Between March 2016 and November 2016, hospitals started with implementation. Data collection started in September 2015 and ended in June 2017 (Supplementary file 1). All patients (aged ≥18 years) admitted for more than 48 hours were eligible for inclusion. Since the study involved a quality improvement intervention with negligible risk of harming patients, individual informed consent was waived for all participating hospitals by the legal department research support of the Amsterdam UMC, location AMC. This trial was registered with the Dutch Trial Registry.

### The discharge process in the Netherlands
In the Netherlands, primary care standards are relatively high and basically, every person has a GP. When a person is hospitalised, responsibility is taken over from the GP by the medical specialist. After discharge, patient care becomes the responsibility of the GP again. It is a policy for hospitals to provide patient handovers to the GP. However, there are no clear guidelines for hospitals how on to arrange their discharge process. The Dutch Health Care Inspectorate[27] indicated that standardised discharge processes are lacking and errors that occur during handovers are often resolved informally.

After discharge from the hospital, the hospital physician sends a medical handover to the primary care provider for every patient (eg, nursing home physician or the GP). Medical handovers include information on the reason for admission, diagnosis, comorbidity, the course of admission, medical examinations, treatment, medication, the health status of the patient at discharge and instructions on follow-up.[28] Nursing handovers are only provided when the patient is discharged to a nursing home or with postdischarge care at the patient's own home. Nursing handovers include information on the care provided during hospitalisation, current nursing care problems, the reason why

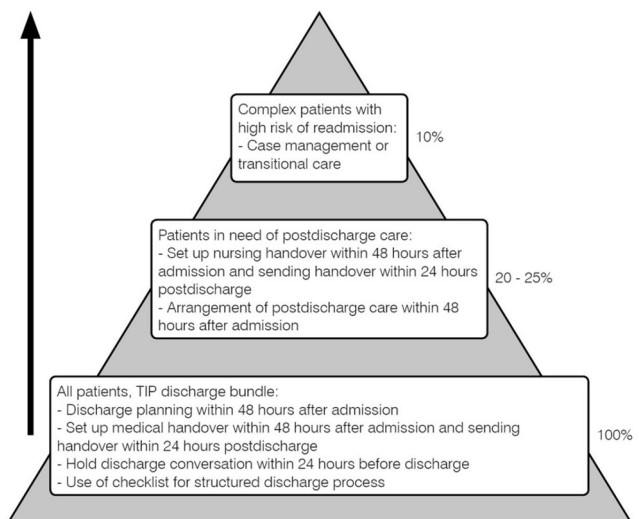

**Figure 1** Pyramid for postdischarge care. A structured discharge process, such as the TIP, procedure should form the basis for every patient. For patients discharged with postdischarge care (20%–25%), nursing handovers should be set up within 48 hours after admission and be sent within 24 hours postdischarge. Complex patients with a high readmission risk (10%) require a (nurse) case manager or transitional care in the transition from hospital to home. TIP, Transfer Intervention Procedure.

(nursing) home care is initiated and the intended outcomes of the care that will be provided.[29]

## Intervention

Figure 1 (adapted from van Seben *et al*[30]) illustrates how the TIP bundle forms the basis of a safe handover from hospital to primary care for every patient, and if applicable, for patients discharged with postdischarge care (eg, home care or a nursing home) or for complex patients who require a case manager or transitional care. As described in two previous studies,[25 31] the TIP bundle was developed using input from focus group meetings with professionals, patient surveys and the literature. The TIP discharge bundle consists of four elements: (1) planning the discharge date within 48 hours after admission and communication of the discharge date with the patient, (2) starting with arrangements for required postdischarge care within 48 hours after admission, (3) preparing patient handovers (medical, medication and nurse) and PPDL[32] within 48 hours after admission and (4) planning a discharge conversation with the patient to explain information from the PPDL 12–24 hours before discharge. The PPDL is a standardised document, containing understandable information for the patient on the reason for admission, hospital treatment, course of the disease, possible sustained consequences or complications and information on medication. We constructed checklists based on the TIP, which served as remembering tool for nurses and physicians in the electronic system or on pocket cards.

## Patient and public involvement

Our research question was developed from the perspective that patients are discharged with little coordination or follow-up and that they are often unprepared at the time of discharge.[4 5] Patients were involved as participants in the construction of the TIP discharge bundle, which was based on, among others, patient satisfaction surveys.[25 31] Further, in a previous study in which the PPDL was developed and implemented, patient satisfaction with the PPDL was also assessed.[32]

## Protocol adherence

To enhance intervention fidelity and protocol adherence in the different hospitals, regular meetings were held with the TIP study group to report results and provide feedback, to discuss implementation, to share experience and to learn from each other's practices. A process evaluation was conducted with the project leaders to investigate protocol adherence, implementation strategies and attention paid to implementation. Elements that were considered included leadership and education of project leaders, projects group, extent of implementation of the discharge bundle and education of physicians and nurses. Feedback points were awarded for all elements and for the extent to which the hospital complied with a certain element, for example, for every person present at the kick-off meeting or for every project meeting that was held. When a hospital partly complied to an element, for example, automatically generated discharge summaries were provided to the patient instead of a PPDL or feedback on timely handovers was only provided to nurses, 0.5 feedback points were awarded. It was not possible to evaluate percentages of compliance with discharge conversations, planning discharge dates and arrangement of postdischarge care within 48 hours since these aspects were not reported in patient records. Hospital policies regarding these elements were assessed.

## Outcome measures

Our primary outcome was the number of medical and nursing handovers sent within 24 hours. This time frame was based on a report of the Dutch Health Care Inspectorate (in Dutch: Inspectie voor de Gezondheidszorg en Jeugd) on the discharge process and handovers, in which it is stated that accurate information needs to be available as quickly as possible, but certainly within 24 hours, for the next care provider.[27] Medical handovers also include medication handovers and we considered the time that these handovers were sent to the GP. The median time between the discharge and the medical handover was considered as secondary outcome. Further, secondary outcomes were LOS and rates of unplanned readmission within 30 days.

## Baseline data collection

Data regarding patient characteristics included demographics, admission ward and medical data (ie, presence of polypharmacy, comorbidity and[33] number of hospitalisation in the 6 months prior to current hospitalisation). Variables were all collected from patient files. All data were reported and analysed anonymously.

## Sample size calculation

On the basis of the findings of a previous study,[31] we expected to find a reduction of 78% in the time between discharge and medical handovers sent. We conducted a power analysis with a number of patients based on the number of hospital beds at the participating wards and feasibility with regards to data collection, which was set at 11 patients. In a simulation study with 16 wards, each contributing 65 patients, we estimated the power to be approximate 91% to demonstrate a reduction of 78% in time until sending the medical handover, assuming that the intraclass correlation coefficient does not exceed 0.05.

## Statistical analysis

Descriptive characteristics of patients were calculated using proportions, means and SD, or medians and IQR, as appropriate. $X^2$ analysis and the Mann-Whitney U test were used to compare preintervention and postintervention patient characteristics. Our time series was divided into two segments, one before and one after implementation of the TIP and we used segmented regression analysis to detect postintervention level changes (ie, an immediate change in the observed outcome after implementation) and changes in postintervention trends relative to preintervention trends (ie, a change in slopes of the regression lines after implementation). A least square regression line was fitted to the two segments of the continuous time variable. The segmented regression helped us to estimate the change in the intercept and the slope coefficients between the preintervention and postintervention period using the following model: $Y_t=\alpha+\beta_1 time_t+\beta_2 intervention_t+\beta_3 time$ after $intervention_t+\varepsilon_t$. Since observations over time are correlated, we explored models with no, a first-order autoregressive correlation between consecutive data collection periods and longer autocorrelation structures.[24] We used the Akaike information criterion (AIC) as an estimator of the relative quality of a model and we report the results from the best fitting model. Correction for baseline imbalances as potential confounders led to results with similar estimates and identical interpretation. On the basis of the extent of protocol adherence and the feedback points awarded, subgroup analyses were performed to assess the intervention effect on the number of medical handovers within 24 hours and the median time between discharge and medical handovers. Statistical analyses were performed using SPSS Statistics V.24.0 and Rstudio V.1.0.136 (Rstudion).

## RESULTS

A total of 2091 patient records (1039 preintervention and 1052 postintervention) were reviewed in order to investigate the effect of the TIP on the timeliness of medical and nursing handovers, LOS and unplanned readmission within 30 days. Overall patients had a mean age (SD) of 68.1 (16.6) years and 46.4% were male (table 1). There were significant differences between the preintervention and postintervention group with regard to polypharmacy and the ratio of acute/elective hospitalisations, and these variables were considered as potential confounders.

However, correction for these potential confounders did not provide better models than the presented models.

## Protocol adherence

Implementation strategies and protocol adherence are summarised in online supplementary file 1. On the basis of the process evaluation, three subgroups were identified. Subgroup 1 (hospitals 4 and 8), >30 feedback points, paid considerable attention to implementation and there was relatively high protocol adherence. In subgroup 2 (hospitals 1–3, and 5), 20–30 feedback points, there was relatively high protocol adherence but moderate attention to implementation. In subgroup 3 (hospitals 6 and 7), <10 feedback points, nearly no attention was brought to implementation and there was low compliance.

## Medical and nursing handovers

In the total study population, no intervention effect was found on the percentage of medical handovers being sent within 24 hours after hospital discharge to the GP: 22.7% medical handovers were sent within 24 hours preintervention, 29.1% postintervention and no significant difference was observed in the levels and trends between the preintervention and postintervention period. The median (IQR) time between discharge and medical handovers decreased from 6.15 (0.96–15.96) days, preintervention, to 4.08 (0.33–13.67) days, postintervention. An absolute effect directly after the implementation of the intervention of −0.25 days was found (ie, the difference in time between discharge and medical handovers between the sixth preintervention data collection point and first postintervention data collection point). We observed no significant difference in the levels and trends. The number of nursing handovers sent within 24 hours postdischarge was 92.8% preintervention and 93.1% postintervention and no significant difference was observed between levels and trends. The results are shown in figure 2 and the parameters estimates are summarised in table 2.

## LOS and unplanned readmission rates

No significant decline in the levels and trends between the preintervention and post-intervention was found with regard to LOS (β 0.08, 95% CI −0.12 to 0.29, p=0.45) and unplanned readmission rates (β 1.11, 95% CI −2.55 to 0.33, p=0.17). Median (IQR) LOS was 8.17 (4.75–15.13) and 8.56 (4.88–15.91) days and readmissions rates were as high as 11.1% and 12.3% preintervention and postintervention, respectively. With regard to LOS, the results are adjusted for autocorrelation (AIC 22.64 vs 33.75, p=0.01), but not for potential confounders (AIC 43.08 vs 33.75, p=0.07). With regard to unplanned readmission rates, the results are unadjusted for autocorrelation (AIC 57.18 vs 54.45, p=0.10) and potential confounders (AIC 57.47 vs 54.45, p=0.61).

## Subgroup analysis

In subgroup 1 (>30 feedback points), an absolute effect of 13.3% more medical handovers sent within 24 hours postdischarge was observed but this did not result in significant changes in level or trends (figure 3). A reduction

**Table 1** Baseline characteristics

| Variable | Overall (n=2091) | Preintervention (n=1039) | Postintervention (n=1052) |
|---|---|---|---|
| Age in years, mean (SD)* | 68.07 (16.57) | 67.66 (16.70) | 68.48 (16.45) |
| Male, n (%) | 971 (46.4) | 493 (47.4) | 478 (45.4) |
| **Living arrangements before admission, n (%)** | | | |
| Independent | 1814 (86.7) | 883 (84.9) | 931 (88.5) |
| Nursing home | 49 (2.3) | 27 (2.6) | 22 (2.1) |
| Senior residence/assisted living | 168 (8.1) | 91 (8.8) | 77 (7.3) |
| Missing | 60 (2.9) | 38 (3.7) | 22 (2.1) |
| **Marital status, n (%)** | | | |
| Married or living together | 1125 (53.8) | 556 (53.5) | 569 (54.1) |
| Single or divorced | 456 (21.8) | 212 (20.4) | 244 (23.2) |
| Widow/widower | 435 (20.8) | 224 (21.6) | 211 (20.1) |
| Missing | 75 (3.6) | 47 (4.5) | 28 (2.7) |
| Charlson Comorbidity Index† (mean, SD*) | 2.05 (2.05) | 2.10 (2.08) | 2.01 (2.03) |
| **Polypharmacy, n (%)‡,§,¶** | 1247 (59.6) | 586 (56.4) | 661 (62.8) |
| Missing | 12 (0.6) | 8 (0.8) | 4 (0.4) |
| Hospitalisation in past 6 months, n (%) | 705 (33.7) | 339 (32.6) | 336 (34.8) |
| Acute hospitalisation, n (%)‡,** | 73.0 (73.0) | 725 (69.8) | 801 (76.1) |
| Admission ward, internal medicine, n (%) | 1051 (50.3) | 524 (50.4) | 527 (50.1) |
| **Discharge destination, n (%)** | | | |
| Home | 1551 (74.2) | 770 (74.1) | 781 (74.2) |
| Other healthcare settings, of which | 482 (23.1) | 238 (23.0) | 244 (23.2) |
| Rehabilitation centre | 268 (12.8) | 120 (11.5) | 148 (14.1) |
| Nursing home | 158 (7.6) | 80 (7.7) | 78 (7.4) |
| Assisted living | 34 (1.6) | 26 (2.5) | 8 (0.8) |
| Another hospital | 22 (1.1) | 12 (1.2) | 10 (1.0) |
| Missing | 58 (2.8) | 31 (3.0) | 27 (2.6) |

*SD.
†Range of 0–31, with a higher score indicating more or more severe comorbidity.[33]
‡Use of five or more different medications.
§$X^2$.
¶P value=0.004.
**P value=0.001.

of 5.6 days in the median time between discharge and handovers with a significant change in level directly after the intervention was observed in subgroup 1 (β −5.29, 95% CI −8.70 to 1.87, p=0.02). Preintervention, group 2 (20–30 feedback points) had the highest rate of medical handovers sent within 24 hours and the lowest median time between discharge and medical handovers but no intervention effect was observed. Both preintervention and postintervention, subgroup 3 (<10 points) had the lowest rates of medical handovers sent within 24 hours, and the highest median time. We observed no intervention effect in subgroup 3.

## DISCUSSION

In the total study population, a structured discharge bundle, the TIP, did not lead to improved timeliness of medical and nursing handovers. Although medical handovers were sent faster postintervention (preintervention median 6.15 days; postintervention median 4.08 days), we were unable to show significant differences in level and trends, both with regard to the median time and the number of medical handovers sent within 24 hours. However, large interhospital variation was observed and a significant intervention effect on the median time between discharge and medical handovers was seen in

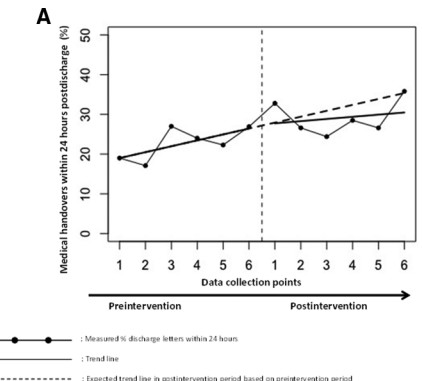
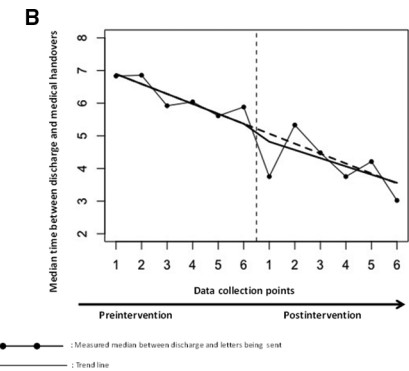

**Figure 2** (A) The number of medical handovers sent within 24 hours. (B) Median time in days between discharge and medical handovers.

those hospitals with relatively high protocol adherence and attention for implementation. Rates of nursing handovers sent within 24 hours both preintervention and postintervention were above 90%. No intervention effect was found for LOS and readmissions.

Extensive research has been conducted to improve patient handovers from hospital to home.[7 16] Summarising findings of earlier discharge interventions that aimed to improve coordination of care and communication between hospital and primary care providers, Hesselink et al[7] and Kripalani et al[8] showed that some studies were able to improve timeliness of medical handovers. These interventions, however, were based on the introduction of fax, email or web-based transfers of information, which is increasingly becoming standard practice in Dutch hospitals. Yet, further improvement may lie in electronic sending systems that support the use of standardised formats that pull information from patient files into (medical) handovers or that send information to the next care provider automatically.

Although a before–after design would probably have led to a significant intervention effect, the ITS analysis provided valuable information on preintervention trends. The observed median time between discharge and sending medical handovers at our first preintervention measurement point was consistent with a recent Dutch study,[14] but a trend towards sending handovers faster was already observed along the preintervention period. During the preintervention period, no interventions were implemented and the TIP was introduced and implemented during a 2-month implementation period during which no measurements were conducted. However, in the preintervention period, attention was already brought to the discharge process, for example, by establishing project groups and the kick-off meeting. Although these activities were not intended as implementation strategies, in hindsight they might explain why improvements were already observed during the preintervention period, particularly since education on the importance of the intervention is an important aspect of implementation.[13 34 35]

Although positive trends in the preintervention period were less pronounced in the subgroup analysis, results of the separate analyses support the idea that attention is important. Whereas a significant reduction of 6 days in the median time between discharge and medical handovers was observed in hospitals that paid much attention to implementation, no intervention effect was observed in hospitals that paid moderate to nearly no attention. It should be noted that the hospitals that paid moderate attention had relatively good preintervention scores. A smaller window for improvement in these hospitals might also explain a lack of intervention effect.[36]

Implementation of the TIP procedure did not reveal a reduction of LOS. Although a possible explanation can be low overall compliance with our study protocol, it is also plausible that over the past years, average LOS has decreased to a minimum.[37] Given current pressure on the availability of hospital beds, patients are discharged as soon as possible. This may account for inadequate discharge processes, since physicians are forced to prioritise acute healthcare over discharge-related tasks.[38 39]

Given increasingly shorter LOS[37] and the often complex care needs a patient face, patient preparation should be an important aspect of the discharge process. In fact, the most effective discharge interventions seem to have educational components.[40] Unfortunately, given the workload among residents, implementation of a PPDL was unsuccessful. For example, posing the question 'do you feel ready to go home'[41] or postdischarge telephone contact[7] might be less time-consuming ways to involve patients. However, to prevent readmissions more effort might be necessary. Previous interventions that revealed a reduction in readmission rates, consist of individualised discharge planning or continue postdischarge.[16 42] However, we believe that a structured discharge process, such as the TIP, should form the basis for a safe handover for every patient (figure 1).

### Implications for further research
Our study shed light on the difficulties that come along with implementation of quality improvement collaboratives.[43] Given the positive preintervention trends and a significant reduction in the median time between discharge and medical handovers in hospitals that paid much attention

van Seben R, et al. BMJ Open 2019;9:e023446. doi:10.1136/bmjopen-2018-023446

**Table 2** Interrupted time series analysis; medical and nursing handovers

| | Medical handovers<24 hours after discharge (%)* | | | Time between discharge and medical letter (days)† | | | Nursing handovers<24 hours after discharge (%)‡ | | |
|---|---|---|---|---|---|---|---|---|---|
| | β (SE) | 95% CI | P value | β (SE) | 95% CI | P value | β (SE) | 95% CI | P value |
| Intercept | 17.51 (3.79) | 10.08 to 24.93 | <0.01 | 7.20 (0.29) | 6.63 to 7.76 | <0.01 | 91.85 (2.71) | 86.53 to 97.16 | <0.01 |
| Trend preintervention (β1) | 1.49 (0.97) | −0.42 to 3.40 | 0.16 | −0.30 (0.07) | −0.45 to −0.16 | <0.01 | 0.28 (0.70) | −1.09 to 1.64 | 0.70 |
| Level change directly after intervention (β2) | 6.43 (10.13) | −13.43 to 26.28 | 0.54 | −0.62 (0.74) | −2.07 to 0.84 | 0.43 | 6.32 (7.25) | −7.89 to 20.53 | 0.41 |
| Trend differences (β3) | −0.94 (1.38) | −3.64 to 1.75 | 0.51 | 0.05 (0.10) | −0.14 to 0.25 | 0.61 | −0.81 (0.99) | −2.74 to 1.12 | 0.43 |
| | Absolute effect directly after intervention: −0.17% | | | Absolute effect directly after intervention: −0.25 days | | | Absolute effect directly after intervention: 0.62% | | |

β1 estimates the preintervention trend.

β2 estimates the difference between the observed level just after the intervention started and that predicted by the preintervention trend.

β3 estimates the difference in trend between the preintervention and postintervention period.

*Correction for autocorrelation did not provide a better model compared with the presented model (AIC 74.17 vs 72.88, p=0.40), nor did correction for potential confounders ('polypharmacy' and 'acute admission') (AIC 74.98 vs 72.88, p=0.39). All models led to results with similar estimates and identical interpretation.

†The results are adjusted for autocorrelation, but not for potential confounders. Correction for autocorrelation (AR1) provided a better model compared with the presented model (AIC 21.52 vs 25.72, p=0.01). Correction for potential confounders ('polypharmacy' and 'acute admission') did not provide a better model compared with the presented model (AIC 29.23 vs 25.72, p=0.78). Correction for autocorrelation (AR1) changed β1 into a significant result. Correction for potential confounders did not alter the results.

‡Correction for autocorrelation did not provide a better model compared with the presented model (AIC 66.05 vs 59.13, p=0.02), nor did correction for potential confounders ('polypharmacy' and 'acute admission') (AIC 59.03 vs 59.13, p=0.13). All models led to results with similar estimates and identical interpretation.

AIC, Akaike information criterion.

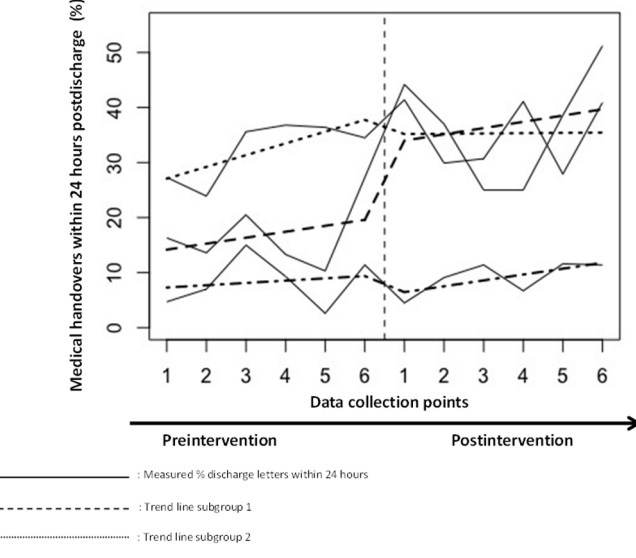

**Figure 3** Hospital differences based on implementation score. The interhospital differences in rates of medical handovers being sent within 24 hours in the preintervention and postintervention based on the extent of implementation and used implementation strategies. Group 1 received >30 feedback implementation points, group 2 received 20–30 feedback implementation points and group 3 received <20 feedback implementation points.

to implementation, further improvements may lie in interventions that create more awareness of the importance of timely handovers and hospital physicians' crucial role in the provision of continuity of care. This might stimulate physicians' intrinsic motivation to provide a structured discharge process and thereby timely handovers.[7][39] Furthermore, we might also want to focus on local factors that lead to insufficient discharge processes. A comprehensive exploration of local barriers for each step in the TIP discharge process might be helpful in order to develop tailor-made interventions on a local or department level to improve the discharge process.[44]

### Limitations

An ITS provides a strong quasi-experimental design to evaluate the impact of an intervention aimed at quality improvement. However, this study design also has limitations. First of all, a positive trend towards sending handovers faster along the preintervention period, which was probably due to the attention that was already brought to the discharge process before implementation of the discharge bundle. In fact, an important limitation of ITS is that it is more difficult to determine whether the observed effect is a direct effect of the intervention, in contrast to, for example, clustered trials. Second, medical staff was not blinded for the outcome measure, that is, timely discharge letters. Knowing that timeliness of discharge letters was monitored might have altered our results. However, in most hospitals, timeliness of discharge letters was already monitored before we started with our research project and the effect is likely to be

minimal. Third, we only recorded the date of sending medical handovers. Knowing whether they were received by GPs would also have provided valuable information. Fourth, we did not look at the content of handovers, while this might have given us important insights. Lastly, it was not possible to evaluate percentages of protocol adherence and the process evaluation with the project leaders might have been an overestimation. However, the process evaluation was in line with the efforts observed during implementation.

### CONCLUSION

Implementation of a structured discharge bundle, the TIP, did not lead to more medical and nursing handovers sent within 24 hours postdischarge. Large interhospital variation was observed, however, and a significant intervention effect on the median time between discharge and medical handovers was seen in those hospitals with high protocol adherence and that brought much attention to implementation. We believe that future interventions should continue to create awareness of the importance of timely handovers and we hope that our study contributes to this, stimulating hospitals to further structure and improve their discharge process.

**Author affiliations**
[1]Department of Internal Medicine, Section of Geriatric Medicine, Amsterdam Public Health research institute, Amsterdam UMC, University of Amsterdam, Amsterdam, The Netherlands
[2]Department of Internal Medicine, Section of Infectious Diseases, Amsterdam UMC, University of Amsterdam, Amsterdam, The Netherlands
[3]Emma Children's Hospital, Amsterdam UMC, University of Amsterdam, Amsterdam, The Netherlands
[4]ACHIEVE Centre of Expertise, Faculty of Health, Amsterdam University of Applied Sciences, Amsterdam, The Netherlands

**Correction notice** This article has been corrected since it first published online. The open access licence type has been amended.

**Collaborators** Hanneke Pullens, Barbara van Munster, Annemarie van der Lugt, Alie Haze-Visser, Agnes van 't Hof, Mariët Dirkzwager, Stella de Regt, Suzan Vroomen, Lisette Bruns.

**Contributors** BMB, SEG and RvS designed the study. BMB and SEG conceived the study and obtained funding. RvS collected data and JMM and RvS performed statistical analysis. All authors contributed to drafting the manuscript or revised it critically and gave final approval for publication. The authors would like to acknowledge the contribution of the TIP study group, consisting in addition to the authors, of the following members Hanneke Pullens (Catharina Hospital Eindhoven), Barbara van Munster (Gelre Hospitals Apeldoorn), Annemarie van der Lugt (Haven Hospital Rotterdam), Alie Haze-Visser (Lange Land Hospital Zoetermeer), Agnes van 't Hof (Maxima Medical Center Veldhoven), Mariët Dirkzwager (OLVG Amsterdam), Stella de Regt (Reinier de Graaf Hospital Delft), Suzan Vroomen (Academic Medical Center Amsterdam) and Lisette Bruns (Dutch Ministry of Health, Welfare and Sport). Furthermore, we also thank Rachel de Vries and Vera van Miltenburg for their assistance with data collection.

**Funding** This work was supported by the Dutch Ministry of Health, Welfare and Sport (grant number: 324798).

**Competing interests** None declared.

**Patient consent for publication** Not required.

**Ethics approval** The Medical Ethics Research Committee (METC) confirmed that the Medical Research Involving Human Subjects Act did not apply to this research project and official approval was not required.

**Provenance and peer review** Not commissioned; externally peer reviewed.

**Data sharing statement** All data relevant to the study are included in the article or uploaded as supplementary information.

**Open access** This is an open access article distributed in accordance with the Creative Commons Attribution 4.0 Unported (CC BY 4.0) license, which permits others to copy, redistribute, remix, transform and build upon this work for any purpose, provided the original work is properly cited, a link to the licence is given, and indication of whether changes were made. See: https://creativecommons.org/licenses/by/4.0/.

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
