## [Reviewer comments · BMJ Open]

ARTICLE DETAILS

TITLE (PROVISIONAL)	A safe handover for every patient: an interrupted time series analysis to test the effect of a structured discharge bundle in Dutch hospitals
AUTHORS	van Seben, Rosanne; Geerlings, Suzanne; Maaskant, Jolanda; Buurman, Bianca

VERSION 1 - REVIEW

REVIEWER	Carl Mahfouz Graduate Medicine, university of Wollongong, Australia
REVIEW RETURNED	30-Apr-2018

GENERAL COMMENTS	Thank you for the opportunity and thanks to the authors for studying such an important topic. The problem of transferring information from hospitals to primary care providers is a worldwide problem to which any clinician anywhere in the world could relate to. Please read my comments as my personal opinion and as merely a constructive feedback and NOT as a criticism! I would like to make general comments first, then some more specific comments related to certain sections of the paper. In general, I thought the terminology regarding the discharge process, which is scattered all over the paper, was a bit confusing- "medical handovers", "nursing handovers", "patient handovers", "discharge summary", "discharge letter", "personalised patient discharge letter (PPDL)", "post discharge care"... "discharge process", 'discharge procedures'.. . As a clinician in Australia with obviously different system to that in the Netherlands, I could not understand the difference. It may be worth simplifying the terminology and/or expanding a bit on the differences, especially between "medical handover", "discharge summaries", "discharge letters" and "PPDL". This may be a specific system in the Netherlands, but it may need clarification as the audience of the article and BMJ open would be world wide. The other general comment/impression is that within the article, there is a lot of mention of various other important aspects of a discharge process that is not very relevant to the question posed and objective of the study. i.e. "accuracy of the discharge information, medication handover, information about change of medications"..., etc.. .
--

Throughout the paper, I noticed minor grammatical and vocabulary errors. It may be worth having someone with English as their first language reviewing the article.

A final point is regarding statistics- as a clinician and I have to admit I personally don't have much experience in statistics, I struggled to understand "levels and slopes" at the beginning of the article. This was explained later in the "statistical analysis" section, but even then, I struggled to understand. The whole "statistical analysis" section may be pitched at researchers, but as a simple clinician, I found it a bit difficult to understand.

Now if I may, I'll comment on specific sections of the article:

Abstract-

The study looked at six pre and post intervention data, yet the stated aim and primary outcome measure was "number of medical and nursing handovers sent within 24h". I am not sure why the 24h? It is well documented in many researches that timely discharge summary is important for various aspects but there was no explanation in the article as to why specifically 24 h was chosen.

page 2 line 22- "within 48 h after admission planning discharge date", may need re-wording?. suggest- planning the discharge date within 48 h after admission .

page 2 line 24-26- "arrangement of post-discharge care, preparing handovers and personalized patient discharge letter"- as someone who doesn't fully understand the Dutch medical system, I don't understand the difference.

Page 2 line 41- "decreased from 6.15 (0.96-15.96) to 4.08 (0.33-13.67)" worth mentioning "days". also "levels and slopes"-

page 3 line 28-30 - "Although it would have been informative, data on the content of medical handovers were not collected, as not on accurateness and timeliness of medication handovers"- suggest a simplified sentence like : Although it would have been informative, data on the content of medical handovers as well as on accuracy and timeliness of medication handovers were not collected.

Introduction-

p4, L7-9 - "Besides, a rising number of older chronically ill patients who move along the care continuum, requires continuity of care" needs re-wording ?

p5, L 22 - typing error, "12 to 24h" (written "tot")

Methods-

p5, L38 - "adhered the SQUIRE.." need to add "to" as in adhered TO the SQUIRE..

Discharge procedures in the Netherlands-

p6, L19- "patient care becomes "the" responsibility of the GP again" (needs "the")

Intervention-

p5-6, first paragraph, I think it needs re-wording. confusing sentence.

p6, L2- "As previously described,25 28 the TIP bundle was developed...", I may have missed it, but I don't believe its mentioned earlier in the article.

Protocol adherence-

p7, L41- "A process evaluation was conducted..." . needs re-wording, "An evaluation process was conducted..."

p7, L 53- "...discharge summaries were provided instead of a PPDL..". I think I mentioned this before, but as an Australian doctor, I don't understand what's the difference between "discharge summary" and "PPDL"?

Outcome measures-

p8, L24-30- "Data regarding patient characteristics included: demographics, admission ward and medical data (i.e. presence of polypharmacy, comorbidity,30 number of hospitalization in the six months prior to current hospitalization). All data were reported and analyzed anonymously". This paragraph seems out of place so to speak- not mentioned anywhere earlier in the article and doesn't seem related to the question posed and primary outcome of "the number of medical and nursing handovers sent within 24h."

Sample size calculation-

I am not sure I understood this section. Unable to comment. It may be something I'm not familiar with..

Statistical analysis-

Unable to comment on this section. As a doctor, not knowing much about statistics, I did not understand it. It may be worth getting a statistician to comment on this section.

Medical and nursing handovers-

p10, L39-41- "medical handovers decreased from 6.15 (0.96-15.96) pre-intervention to 4.08 (0.33-13.67) post-intervention" .I'm guessing these are days. Worth adding "days"..

Subgroup analysis-

p11, L28-32- "...and the lowest median time between discharge and medical handovers but no intervention effect was". incomplete sentence. I believe it needs re-phrasing.

Discussion-

p12, L28-41- "While the observed median time between discharge and sending medical handovers at our first pre-intervention measurement point was consistent with a recent Dutch study, a trend towards sending handovers faster was observed along the pre-intervention period. During this period, no interventions were implemented but attention was already brought to the discharge procedure, e.g. by establishing project groups and the kick-off meeting. Since education on the importance of the intervention is an important aspect of implementation, this could explain why improvements were already observed." Reading this section, I could only make an assumption that there was an overlap between the collection of data pre "TIP" and the implementation of the "TIP" which is not mentioned/explained in the paper. Also, if this was the case, this would compromise the validity of the study as it makes it very hard to analyse pre and post "TIP" data accurately. (my personal opinion)

	I hope these comments are some help to the authors. Thank you.
--	---

REVIEWER	Beth Fylan Bradford Institute for Health Research / University of Bradford, UK
REVIEW RETURNED	01-May-2018

GENERAL COMMENTS	This is an interesting study aimed at evaluating an intervention to improve transitions of care. Having reviewed the manuscript and the protocol, there are a few issues the authors could address:  • The patient consent process is not clear. How were patients consented into the study? The team accessed their data so there must have been a consent process. • It seems the rates of timely nursing handover pre-interventions were high. Were data collected about sites and their performance in timely handover prior to launching the study? • The authors describe a complex multi-component intervention but the primary outcome measure seemingly targeting enhanced information and preparation for patients as well as better organisation of discharge. The primary outcome just measures time taken to send discharge information. • Could there be an effect of staff knowing they were being timed on performance? Were they aware time to send handover information was being recorded during the pre-intervention period and could this have had an impact on the time taken to send discharge information? • The protocol includes some qualitative work which is not reported here. Is it going to be reported elsewhere? • Do the authors have a comment about the how medical and nursing handovers differ? It is not clear what the difference between these two handovers is. Is there more detail required in medical handovers? • Can they comment on the completeness of the checklists referenced in the study protocol? To what extent were these completed, collected and reviewed? The authors state they are unable to provide information about adherence to discharge conversations, discharge date planning and follow-up care. Would the checklists have done this? • A comment about the steps taken to enhance intervention fidelity and adherence would be useful, given the large inter-hospital variation. Specific points  • In the strengths and limitations bullet points, the final two statements are not clear. • Introduction section: it is not clear what you mean by “Besides, a rising number of older chronically ill patients who move along the care continuum, requires continuity of care.”
---

	• In the PPI section it is not clear if patients were involved in the study team or just as participants in developing the intervention. Other recent refs: Waring, J.; Bishop, S.; Marshall, F.; (2016), "A qualitative study of professional and carer perceptions of the threats to safe hospital discharge for stroke and hip fracture patients in the English National Health Service", BMC Health Services Research, Vol.16, 296. Rustad EC, Furnes B, Cronfalk BS, et al. Older patients' experiences during care transition. Patient Prefer Adherence 2016;10:769–79.
--	---

REVIEWER	PE Waterson Loughborough University, UK
REVIEW RETURNED	20-Jun-2018

GENERAL COMMENTS	This struck me as a well-written and executed study. The quality of the study is to be commended. At the same time I found this is a rather puzzling paper. The authors report a failure to find any effect of a structured discharge bundle on the timeliness of patient handovers. Perhaps that alone is sufficiently interesting for the paper to be published by BMJ Open. However, I doubt that this is strong enough to warrant publications alone. I would have expected that a negative result would mean that the authors would argue for more scepticism towards the impact of these types of bundles. That didn't come through in the paper. The paper reminded me of one of those 'failure to replicate' papers that you come across occasionally – sometime these are interesting and worthwhile (e.g., debunking an assumption or throwing doubt an important earlier finding), most times they are a little soulless. I also don't think that a paper should be published (at least not in BMJ Open) which states that the 'design of the study was a strength' (page 14, line 11) – that isn't really strong enough. The design is good, but what is the real message here. Should we be rethinking these types of bundles? Should we be doing something else? I couldn't really find much in the way of guidance for the future in the paper.
---

VERSION 1 – AUTHOR RESPONSE

Reviewer: 1 - Reviewer Name: Carl Mahfouz

1. In general, I thought the terminology regarding the discharge process, which is scattered all over the paper, was a bit confusing- "medical handovers", "nursing handovers", "patient handovers", "discharge summary", "discharge letter", "personalised patient discharge letter (PPDL)", "post discharge care"... "discharge process", 'discharge procedures'.. . As a clinician in Australia with obviously different system to that in the Netherlands, I could not understand the difference. It may be worth simplifying the terminology and/or expanding a bit on the differences, especially between

"medical handover", "discharge summaries", "discharge letters" and "PPDL". This may be a specific system in the Netherlands, but it may need clarification as the audience of the article and BMJ open would be world wide.

Author response:

We thank the reviewer for addressing this issue and apologize for any confusion. We use more consistent language now throughout the paper:

A. We removed the term 'discharge procedure' and consistently refer to 'discharge process' now;

B. We avoid the words 'discharge letter' and refer to medical and nursing handovers throughout the manuscript;

C. We avoid the term 'discharge summary', where we were actually describing medical handovers. Only in our method section we do refer to actual discharge summaries, since these were provided to patients in some of the hospitals instead of the personalized patient discharge letter (PPDL). But these are defined more clearly now:

"When a hospital partly complied to an element, e.g. automatically generated discharge summaries were provided to the patient instead of a PPDL or feedback on timely handovers was only provided to nurses, 0.5 feedback points were awarded."

Further, we define specific terms when they are first introduced, specifically:

D. We define 'medical handover' by stating:

"The root of a safe transition from hospital to home or nursing home is a timely transfer of the medical handover, that is a letter containing accurate medical discharge information for the following care provider^{8 13}."

E. We define 'post-discharge care' in our method section now by stating:

"For patients discharged with post-discharge care (e.g., home care or a nursing home) (...)"

F. We provide more clarification on the personalized patient discharge letter (PPDL). In the introduction we rephrased the term into 'personalized discharge letter for the patient (PPDL)' to provide more clarification:

In the method section we provide a more clear definition of the PPDL by stating:

"The PPDL is a standardized document, containing understandable information for the patient on the reason for admission, hospital treatment, course of the disease, possible sustained consequences or complications, and information on medication."

2. The other general comment/impression is that within the article, there is a lot of mention of various other important aspects of a discharge process that is not very relevant to the question posed and objective of the study. i.e. "accuracy of the discharge information, medication handover, information about change of medications"..., etc.. .

Author response:

We agree with the reviewer and hindsight we see that, at times, we elaborated on information that we did not seek to address in our study. Therefore, we only describe the importance of timely discharge information in our introduction section now, and do not refer to accuracy of discharge information. Furthermore, we have removed the information on 'medication handovers and changes of medication'

from our limitation section and from the bullet points with strengths and limitations of this study. We wrote a new bullet point instead, which reads as:

“Only the date of sending patient handovers were recorded. Knowing whether the next care provider received information would have been informative.”

3. Throughout the paper, I noticed minor grammatical and vocabulary errors. It may be worth having someone with English as their first language reviewing the article.

Author response:

We thank the reviewer for the careful reading. We believe the suggestions below have helped to resolve most of the grammatical and vocabulary errors. Further, we have carefully reviewed and edited our article.

4. A final point is regarding statistics- as a clinician and I have to admit I personally don't have much experience in statistics, I struggled to understand "levels and slopes" at the beginning of the article. This was explained later in the "statistical analysis" section, but even then, I struggled to understand. The whole "statistical analysis" section may be pitched at researchers, but as a simple clinician, I found it a bit difficult to understand. Now if I may, I'll comment on specific sections of the article:

A. Abstract - The study looked at six pre and post intervention data, yet the stated aim and primary outcome measure was "number of medical and nursing handovers sent within 24h". I am not sure why the 24h? It is well documented in many researches that timely discharge summary is important for various aspects but there was no explanation in the article as to why specifically 24 h was chosen.

Author response:

We added information on the chosen time frame in the method section under the heading outcome measures by stating:

“This time-frame was based on a report of the Dutch healthcare inspectorate (In Dutch: Inspectie voor de Gezondheidszorg en Jeugd (IGJ)) on patient handovers, in which it is stated that accurate information needs to be available as quick as possible, but certainly within 24 hours for the following care provider.¹”

B. page 2 line 22- "within 48 h after admission planning discharge date", may need re-wording?. suggest- planning the discharge date within 48 h after admission.

Author response:

We have re-written this sentence in the abstract and throughout the manuscript.

C. page 2 line 24-26- "arrangement of post-discharge care, preparing handovers and personalized patient discharge letter"- as someone who doesn't fully understand the Dutch medical system, I don't understand the difference.

Author response:

We understand that it might be difficult for readers to understand the difference between post-discharge care, preparing handovers and a personalized patient discharge letter. Given the limited amount of words that can be used in the abstract we are unable to elaborate on the differences in the abstract. But we do provide more clarification in the introduction and method section. This also follows up on comment 1 of the reviewer (see our answer at comment 1).

D. Page 2 line 41- "decreased from 6.15 (0.96-15.96) to 4.08 (0.33-13.67)" worth mentioning "days". also "levels and slopes"-

Author response:

We have added the word days.

E. Page3 line 28-30 - "Although it would have been informative, data on the content of medical handovers were not collected, as not on accurateness and timeliness of medication handovers"- suggest a simplified sentence like : Although it would have been informative, data on the content of medical handovers as well as on accuracy and timeliness of medication handovers were not collected.

Author response:

We thank the reviewer for this suggestion. However, based on comment 2 of the reviewer we have removed the sentence on accuracy of medication handovers.

F. Introduction-

p4, L7-9 - "Besides, a rising number of older chronically ill patients who move along the care continuum, requires continuity of care" needs re-wording ? p5, L 22 - typing error, "12 to 24h" (written "tot")

Author response:

We have changed tot into "to" and we have altered the sentence into:

"Besides, a rising number of older chronically ill patients who move within the health care system, requires continuity of care".

G. Methods - p5, L38 - "adhered the SQUIRE.." need to add "to" as in adhered TO the SQUIRE..

Author response:

We have added to.

H. Discharge procedures in the Netherlands- p6, L19- "patient care becomes "the" responsibility of the GP again" (needs "the")

Author response:

We have added 'the'.

I. Intervention-

p5-6, first paragraph, I think it needs re-wording. confusing sentence.

p6, L2- "As previously described,25 28 the TIP bundle was developed...", I may have missed it, but I don't believe its mentioned earlier in the article.

Author response:

"As previously described" refers to our study protocol and another previous study on the TIP bundle. We have re-phrased the sentence into:

"As described in two previous studies,25,31 (...)

J. Protocol adherence-

p7, L41- "A process evaluation was conducted..." . needs re-wording, "An evaluation process was conducted..."

Author response:

We correctly refer to a 'process evaluation' as a concept, see for example 'Process evaluation in randomized controlled trials of complex interventions' Oakley et al., BMJ 2006

K. p7, L 53- "...discharge summaries were provided instead of a PPDL..". I think I mentioned this before, but as an Australian doctor, I don't understand what's the difference between "discharge summary" and "PPDL"?

Author response:

Thank you for pointing out that it was unclear what the difference was between discharge summaries and PPDLs. This comment also follows up on comment 1 by the reviewer (see answer section 1F).

L. Outcome measures-

p8, L24-30- "Data regarding patient characteristics included: demographics, admission ward and medical data (i.e. presence of polypharmacy, comorbidity,30 number of hospitalization in the six months prior to current hospitalization). All data were reported and analyzed anonymously". This paragraph seems out of place so to speak- not mentioned anywhere earlier in the article and doesn't seem related to the question posed and primary outcome of "the number of medical and nursing handovers sent within 24h."

Author response:

We agree with the reviewer that this information is not directly related to our outcome measures, instead it refers to baseline data collection, for which we wrote a separate paragraph now in our method section: "Baseline data collection".

M. Sample size calculation-

I am not sure I understood this section. Unable to comment. It may be something I'm not familiar with.. Statistical analysis- Unable to comment on this section. As a doctor, not knowing much about statistics, I did not understand it. It may be worth getting a statistician to comment on this section.

Author response:

A statistician was consulted and conducted the sample size calculation.

N. Medical and nursing handovers-

p10, L39-41- "medical handovers decreased from 6.15 (0.96-15.96) pre-intervention to 4.08 (0.33-13.67) post-intervention" .I'm guessing these are days. Worth adding "days"..

Author response:

We have added the word 'days'.

O. Subgroup analysis-

p11, L28-32- "...and the lowest median time between discharge and medical handovers but no intervention effect was". incomplete sentence. I believe it needs re-phrasing.

Author response:

We thank the reviewer for the careful reading, we have added the word observed at the end of the sentence.

P. Discussion-

p12, L28-41- "While the observed median time between discharge and sending medical handovers at our first pre-intervention measurement point was consistent with a recent Dutch study, a trend towards sending handovers faster was observed along the pre-intervention period. During this period, no interventions were implemented but attention was already brought to the discharge procedure, e.g. by establishing project groups and the kick-off meeting. Since education on the importance of the intervention is an important aspect of implementation, this could explain why improvements were already observed."

Reading this section, I could only make an assumption that there was an overlap between the collection of data pre "TIP" and the implementation of the "TIP" which is not mentioned/explained in the paper. Also, if this was the case, this would compromise the validity of the study as it makes it very hard to analyse pre and post "TIP" data accurately. (my personal opinion)

Author response:

We thank the reviewer for pointing to this issue. It is important to note that there was no overlap between the collection of the pre-intervention data and implementation of the TIP. Instead, there were six pre-intervention measurements, which was followed by an implementation period of two months, during which no measurements were conducted. After implementation, six post-intervention measurements were conducted. To avoid any confusion, we provide more clarification on this in the method section now, by stating:

"Outcomes before and after implementation of the TIP bundle were compared. Therefore, six pre-intervention measurements were conducted before implementation of the TIP and six post-intervention measurements after implementation. During the implementation period of two months no measurements were conducted."

That said, we were also surprised by the positive trend observed during the pre-intervention period. However, although the TIP was not actually implemented during the pre-intervention period, attention was brought to the discharge process. Hence, the positive trend observed during the pre-intervention period, is potentially explained by the fact that we created awareness on the importance of a solid discharge procedure already during the pre-intervention period. It seemed that, in hindsight, it might have been an important element that helped improving the discharge process. In our discussion we provide more clarification on this now by stating:

"Although a before-after design would probably have led to a significant intervention-effect, the ITS analysis provided valuable information on pre-intervention trends. The observed median time between discharge and sending medical handovers at our first pre-intervention measurement point was consistent with a recent Dutch study¹⁴, but a trend towards sending handovers faster was already observed along the pre-intervention period. During the pre-intervention period, no interventions were implemented and the TIP was introduced and implemented during a two-month implementation period during which no measurements were conducted. However, in the pre-intervention period, attention was already brought to the discharge process, e.g. by establishing project groups and the kick-off meeting. Although these activities were not intended as implementation strategies, in hindsight they might explain why improvements were already observed during the pre-intervention period, particularly since education on the importance of the intervention is an important aspect of implementation^{13 33 34}."

Reviewer: 2 Reviewer Name: Beth Fylan

5. The patient consent process is not clear. How were patients consented into the study? The team accessed their data so there must have been a consent process.

Author response:

We thank the reviewer for pointing to the lack of clarification on the patient consent process. Since patients received care as usual and this study involved a quality improvement project, individual informed consent was waived for all participating hospitals. We have added this information in the method section of our manuscript, paragraph 'study design and setting', where we now state:

"Since the study involved a quality improvement intervention with negligible risk of harming patients, individual informed consent was waived for all participating hospitals".

6. It seems the rates of timely nursing handover pre-interventions were high. Were data collected about sites and their performance in timely handover prior to launching the study?

Author response:

Data on nursing handovers were indeed collected prior to launching the study. This process was similar as for the process with regard to medical handovers (see answer Comment 4P). However, with regard to the nursing handovers, no positive trend was observed during the pre-intervention period, but rates of timely nursing handovers were indeed high both along the total pre-intervention and post-intervention period.

7. The authors describe a complex multi-component intervention but the primary outcome measure seemingly targeting enhanced information and preparation for patients as well as better organisation of discharge. The primary outcome just measures time taken to send discharge information.

Author response:

Given that a safe transition from hospital to home starts with a timely transfer of the patient handover from hospital to the primary care provider, we decided to measure the intervention-effect on a timely handover. As the reviewer correctly addresses, our intervention seemed to have had the potential to address also other outcomes related to the discharge process. Therefore, length of hospital stay and unplanned hospital readmissions were defined as secondary outcome measures.

8. Could there be an effect of staff knowing they were being timed on performance? Were they aware time to send handover information was being recorded during the pre-intervention period and could this have had an impact on the time taken to send discharge information?

Author response:

Staff were indeed aware that the time taken to send discharge information was monitored. However, in most hospitals discharge letters were already monitored before we started with our study and the effect on our study results is probably minimal. However, we agree with the reviewer that this should be mentioned and we report on this now in our limitation section, by stating:

"Medical staff was not blinded for the outcome measure, that is timely discharge letters. Knowing that timeliness of discharge letters was monitored might have altered our results. However, in most hospitals timeliness of discharge letters was already monitored before we started with our research project and the effect is likely to be minimal."

9. The protocol includes some qualitative work which is not reported here. Is it going to be reported elsewhere?

Author response:

We thank the reviewer for the careful reading of our study protocol. Partially, qualitative work has resulted in the process evaluation as reported in its current form. Additionally, we have prepared a separate manuscript of the qualitative work that we conducted.

10. Do the authors have a comment about the how medical and nursing handovers differ? It is not clear what the difference between these two handovers is. Is there more detail required in medical handovers?

Author response:

Thank you for pointing out this lack of clarification. In the method section, under the heading discharge process in the Netherlands, we elaborate on the differences between medical and nursing handover now by stating:

“After discharge from the hospital medical handovers, the hospital physician sends a medical handover to the primary care provider for every patient (e.g., nursing home physician or the GP). Medical handovers include information on the reason for admission, diagnosis, comorbidity, the course of admission, medical examinations, treatment, medication, the health status of the patient at discharge, and instructions on follow-up²⁸. Nursing handovers are only provided when the patient is discharged to a nursing home or discharged home with post-discharge care at home. Nursing handovers include information on the care provided during hospitalization, current nursing care problems, the reason why (nursing) home care is initiated, and the intended outcomes of the care that will be provided²⁹.

11. Can they comment on the completeness of the checklists referenced in the study protocol? To what extent were these completed, collected and reviewed? The authors state they are unable to provide information about adherence to discharge conversations, discharge date planning and follow-up care. Would the checklists have done this?

Author response:

The checklist, containing all elements of the TIP, was used as remembering tool, which was available in the electronic system or on pocket cards. We provide more clarification on the purpose of the checklist in the method section, under the heading ‘intervention’, by stating:

“We constructed checklists based on the TIP, which served as remembering tool for nurses and physicians in the electronic system or on pocket cards. “

Checklists were thus not used an instrument to measure the actual care provided and indeed we were unable to collect data on percentages of compliance. This is a limitation of our study, which we address in our limitation section, by stating:

“Lastly, it was not possible to evaluate percentages of protocol adherence and the process evaluation with the project leaders might have been an overestimation.”

12. A comment about the steps taken to enhance intervention fidelity and adherence would be useful, given the large inter-hospital variation.

Author response:

A TIP study group was established, comprising a study coordinator, two supervisors, one clinical epidemiologist, a policy officer from the Ministry of VWS and local project leaders from the eight participating Dutch hospitals that implemented the TIP bundle. We stated in the method section that regular meetings were held with the study group. However, due to the process of cumulating words

we did not elaborate on the purpose of these meetings, which were, in fact, held to enhance intervention fidelity and protocol adherence in the different hospitals. We elaborate on this now in our method section, under the heading 'Protocol adherence', by stating:

"To enhance intervention fidelity and protocol adherence in the different hospitals, regular meetings were held with the TIP study group to report results and provide feedback, to discuss implementation, share experience and learn from each other's practices."

13. Specific points

A. In the strengths and limitations bullet points, the final two statements are not clear.

Author response:

Based on comment 2 of reviewer 1, we removed the second last statement. We have altered the final statement, and this sentence now reads as:

"It was not possible to evaluate percentages of compliance with the study protocol. Therefore, the process evaluation with the project leaders might have been an overestimation."

B. Introduction section: it is not clear what you mean by "Besides, a rising number of older chronically ill patients who move along the care continuum, requires continuity of care."

Author response:

This comment follows up on comment 4F of reviewer 1 and we have altered the sentence (see answer 4F).

C. In the PPI section it is not clear if patients were involved in the study team or just as participants in developing the intervention.

Author response:

Patients were involved as participants in developing in the intervention, which we clarify now by stating:

"Our research question was developed from the perspective that patients are discharged with little coordination or follow-up and that they are often unprepared at time of discharge^{4 5}. Patients were involved as participants in the construction of the TIP discharge bundle, which was based on, among others, patient satisfaction surveys^{25 31}. Further, in a previous study in which the PPDL was developed and implemented, patient satisfaction with the PPDL was also assessed³²."

Reviewer: 3 - Reviewer Name: PE Waterson

14. This struck me as a well-written and executed study. The quality of the study is to be commended. At the same time I found this is a rather puzzling paper. The authors report a failure to find any effect of a structured discharge bundle on the timeliness of patient handovers. Perhaps that alone is sufficiently interesting for the paper to be published by BMJ Open. However, I doubt that this is strong enough to warrant publications alone. I would have expected that a negative result would mean that the authors would argue for more scepticism towards the impact of these types of bundles. That didn't come through in the paper. The paper reminded me of one of those 'failure to replicate' papers that you come across occasionally – sometime these are interesting and worthwhile (e.g., debunking an assumption or throwing doubt an important earlier finding), most times they are a little soulless. I also don't think that a paper should be published (at least not in BMJ Open) which states that the 'design

of the study was a strength' (page 14, line 11) – that isn't really strong enough. The design is good, but what is the real message here. Should we be rethinking these types of bundles? Should we be doing something else? I couldn't really find much in the way of guidance for the future in the paper.

Author response:

We thank the reviewer for the kind words on the quality of our study and we are sensitive to the concerns raised. Based on the comments of the reviewer we made adjustments of which we hope that they will strengthen the message of our paper.

The reviewer points out that more skepticism towards discharge bundles might be appropriate, as indeed a negative effect was observed. We believe that the lack of intervention-effect is mostly explained by low overall compliance and that our study shed light on the difficulties that come along with implementation of quality improvement collaboratives (Hulscher et al., 2013, *BMJ Quality & Safety* (42)). A large inter-hospital variation was observed and our study showed that effort and attention for the importance of the intervention pay off. Therefore, we believe that future interventions should focus on awareness creation of hospital physicians' crucial role in the provision of continuity of care. We have altered the second paragraph of our discussion section to strengthen this message.

During the pre-intervention period, no interventions were implemented and the TIP was introduced and implemented during a two-month implementation period during which no measurements were conducted. However, in the pre-intervention period, attention was already brought to the discharge process, e.g. by establishing project groups and the kick-off meeting. Although these activities were not intended as implementation strategies, in hindsight they might explain why improvements were already observed during the pre-intervention period, particularly since education on the importance of the intervention is an important aspect of implementation^{13 33 34}.

Further we elaborate on this in our 'implications for further research' paragraph:

"Our study shed light on the difficulties that come along with implementation of quality improvement collaboratives⁴². Given the positive pre-intervention trends and significant reduction in the median time between discharge and medical handovers in hospitals that paid much attention to implementation, our study showed that effort pays off. Therefore, further improvements may lie in interventions that create more awareness of the importance of timely handovers and hospital physicians' crucial role in the provision of continuity of care. This might stimulate physicians' intrinsic motivation to provide a structured discharge process and thereby timely handovers^{7 38}."

In addition, given the large inter-hospital variation, we believe that we should focus on tailor-made interventions to improve the discharge process. We have further altered our paragraph on implications for further research to strengthen this message:

"Furthermore, we might also want to focus on local factors that lead to insufficient discharge processes. A comprehensive exploration of local barriers for each step in the TIP discharge process might be helpful in order to develop tailor made interventions on a local or department level to improve the discharge process⁴³."

Lastly, the reviewer addresses that the design of the study cannot be considered as a strength. Yet, the ITS analysis provided valuable information on pre-intervention trends, and we moved this information from our strength and limitation section to the second paragraph of our discussion, where we now state:

"Although a before-after design would probably have led to a significant intervention-effect, the ITS analysis provided valuable information on pre-intervention trends. The observed median time between discharge and sending medical handovers at our first pre-intervention measurement point was

consistent with a recent Dutch study¹⁴, but a trend towards sending handovers faster was already observed along the pre-intervention period. (...).”

VERSION 2 – REVIEW

REVIEWER	Carl Mahfouz Graduate Medicine, University of Wollongong Australia
REVIEW RETURNED	24-Sep-2018

GENERAL COMMENTS	Thank you again for the very important work you are undertaking. I can notice a huge improvement on the first version of this manuscript. I only have few points to outline and apologies if my comments are not very relevant and/or if I missed any clarifications or answers to them somehow... There is mention of "six' pre and post intervention points.. yet the outcomes measured where "four" points-1- number of handovers in 24h, 2- median time between discharge and handovers, 3- length of hospital stay and 4- unplanned readmissions .?? I don't think I read those six points anywhere in the paper . Page 6, line 51- "After discharge from the hospital medical handovers, the hospital physician sends....." needs rephrasing. Sample size calculation section - I struggled to understand it. Statistical analysis- I am non expert on statistics but reading this section, I found it hard to understand -- " Chi-squared analysis", "Mann Whitney test", "generalised least square analysis", "levels and slopes" (BTW, levels and slopes is used a lot all through the article from the abstract on page 2 to page 12 where its used 3 times in various sections- may be worth explaining for the statistically challenged doctors..???) .."We explored models with no, a first order autoregressive correlation between consecutive data collection periods, and longer autocorrelation structures." "We used the Akaike Information Criterion (AIC) as an estimator.." I am not sure if it needs to or how this section can be simplified so it can be understood by your average non statistician reader? In the Medical and nursing handovers section- it says "The median (interquartile range, IQR) time between discharge and medical handovers decreased from 6.15 (0.96-15.96) days, pre-intervention to 4.08 (0.33-13.67) days post-intervention. An absolute effect directly after the implementation of the intervention of -0.25 days was found." - I am not sure I understand where the -0.25 days came from..?? It looks like the time was reduced by a good 2 days from 6.15 pre to 4.08 days post implementation ??? Thank you
--

REVIEWER	Beth Fylan Bradford Teaching Hospitals NHS Foundation Trust; University of Bradford.
REVIEW RETURNED	04-Oct-2018

GENERAL COMMENTS	1. The amendments to the paper have improved the reader's understanding of medical and nursing handovers. I am still concerned about why the primary outcome measure is timeliness of communication when the bundle aims to improve many aspects of care presumably with the aim of enhancing a patient outcome, for example reducing re-admission, reducing post-discharge deterioration or enhancing patient knowledge of their condition and post-discharge plan. The secondary outcome measure of unplanned readmissions seems to be a better primary outcome. 2. The problems with high time measurements before implementation due to increased pre-intervention focus on transferring information quickly should be noted in the limitations. 3. In the abstract the sentence "No significant change in levels and slopes was observed in the number of medical and nursing handovers sent within 24h" still will not make sense to people who don't understand what levels and slopes means. Can you just say: 'No significant change was observed in the number of medical and nursing handovers sent within 24h.' 4. There are occasional typos / grammar issues that should be addressed: e.g. 'Nonetheless, a review of Kripalani et al. showed that medical handover are often not available...' 5. You say that 'Since the study involved a quality improvement intervention with negligible risk of harming patients, individual informed consent was waived for all participating hospitals.' Can you specify who waived this consent? 6. A comment about the benefits and drawbacks of measuring the effect of complex interventions through this type of design would be useful with a comparison to other designs which might be more expensive and time consuming – such as cluster trials.
---

REVIEWER	Julio Díaz Carlos III Institute of Health.
REVIEW RETURNED	11-Jan-2019

GENERAL COMMENTS	From the statistical point of view, the methodology used is correct. However, the following aspects should be explained in more detail: 1. The equations of the adjustment lines must be shown by the method of least squares and indicate the statistical significance of their coefficients, especially the trend. 2. The authors on page 10 indicate that they have explored models that have compared with the Akaike Information Criterion. However, these models are not shown in results. This aspect must be clarified and the results shown.
--

VERSION 2 – AUTHOR RESPONSE

Reviewer: 1 Carl Mahfouz

Thank you again for the very important work you are undertaking. I can notice a huge improvement on the first version of this manuscript. I only have few points to outline and apologies if my comments are not very relevant and/or if I missed any clarifications or answers to them somehow...

1. There is mention of 'six' pre and post intervention points.. yet the outcomes measured where "four" points-1- number of handovers in 24h, 2- median time between discharge and handovers, 3- length of hospital stay and 4- unplanned readmissions .?? I don't think I read those six points anywhere in the paper .

Author response

We apologize for any confusion. An interrupted time series involves repeated observations of a particular event collected over time, divided into two segments in our case. Hence, six pre- and post-intervention points refers to the number of repeated measurements that were conducted. At these in total 12 data collection point, our outcome measures were measured. We have clarified this throughout the paper. In the abstract we state:

“Interrupted time series with six pre-intervention and six post-intervention data collection points (...).”

In our method section, paragraph study design and setting we now state:

“Within an interrupted time series, repeated observations are collected over time and divided into two segments, one before and one after implementation. Therefore, at six pre-intervention data collection points, measurements were conducted before implementation of the TIP and at six post-intervention data collection points measurements were conducted after implementation.”

In addition, the six data collection points pre-intervention and post-intervention are also shown in figure 2a, 2b, and 3.

2. Page 6, line 51- "After discharge from the hospital medical handovers, the hospital physician sends....." needs rephrasing.

Author response

We have rephrased the sentence into:

“After discharge from the hospital, the hospital physician sends a medical handover to the primary care provider for every patient (e.g., nursing home physician or the GP).”

3. Sample size calculation section - I struggled to understand it.

Statistical analysis- I am non expert on statistics but reading this section, I found it hard to understand -- " Chi-squared analysis", "Mann Whitney test", "generalised least square analysis", "levels and slopes" (BTW, levels and slopes is used a lot all through the article from the abstract on page 2 to page 12 where its used 3 times in various sections- may be worth explaining for the statistically challenged doctors..???) .."We explored models with no, a first order autoregressive correlation between consecutive data collection periods, and longer autocorrelation structures." "We used the Akaike Information Criterion (AIC) as an estimator.." I am not sure if it needs to or how this section can be simplified so it can be understood by your average non statistician reader?

Author response

We are sorry to hear that the reviewer experienced difficulties with understanding our statistical analysis. We tried to be as clear as possible and we are of the opinion that explaining commonly used statistical tests as the Chi-squared analysis and Mann Withney goes beyond the scope of our paper. Explaining statistical analyses to a non-statistician reader is challenging and we would like recommend the study of Penfold et al, to which we refer in our paragraph on analysis. Yet, to be as clear as possible, we have re-written our paragraph on statistical analyses and provide more explanation on our segmented regression analyses. Also, we have replaced 'slope' with trend throughout the paper, which is hopefully more clear. Our paragraph on statistical analyses now reads as:

"Our time series was divided into two segments, one before and one after implementation of the TIP and we used segmented regression analysis to detect post-intervention level changes (i.e., an immediate change in the observed outcome after implementation) and changes in post-intervention trends relative to pre-intervention trends (i.e., a change in slopes of the regression lines after implementation). A least square regression line was fitted to the two segments of the continuous time variable. The segmented regression helped us to estimate the change in the intercept and the slope coefficients between the pre-intervention and post-intervention period using the following model: $Y_t = \alpha + \beta_1 \text{time}_t + \beta_2 \text{intervention}_t + \beta_3 \text{time after intervention}_t + \epsilon_t$. Since observations over time are correlated, we explored models with no, a first order autoregressive correlation between consecutive data collection periods, and longer autocorrelation structures.¹ We used the Akaike Information Criterion (AIC) as an estimator of the relative quality of a model and we report the results from the best fitting model. Correction for baseline imbalances as potential confounders led to results with similar estimates and identical interpretation."

1. Penfold RB, Zhang F. Use of interrupted time series analysis in evaluating health care quality improvements. *Academic pediatrics* 2013;13(6 Suppl):S38-44. doi: 10.1016/j.acap.2013.08.002 [published Online First: 2013/12/07]

4. In the Medical and nursing handovers section- it says "The median (interquartile range, IQR) time between discharge and medical handovers decreased from 6.15 (0.96-15.96) days, pre-intervention to 4.08 (0.33-13.67) days post-intervention. An absolute effect directly after the implementation of the intervention of -0.25 days was found." - I am not sure I understand where the -0.25 days came from..?? It looks like the time was reduced by a good 2 days from 6.15 pre to 4.08 days post implementation???

Author response

The median time between pre and post-implementation indeed decreased with 2 days. A direct intervention effect refers to the -.025 between the last pre-intervention data collection point and first post-intervention collection point. We provide more clarification by stating:

"An absolute effect directly after the implementation of the intervention of -0.25 days was found (i.e., de difference in time between discharge and medical handovers between the sixth pre-intervention data collection point and first post-intervention data collection point)."

Reviewer: 2 Beth Fylan

5. The amendments to the paper have improved the reader's understanding of medical and nursing handovers. I am still concerned about why the primary outcome measure is timeliness of communication when the bundle aims to improve many aspects of care presumably with the aim of enhancing a patient outcome, for example reducing re-admission, reducing post-discharge

deterioration or enhancing patient knowledge of their condition and post-discharge plan. The secondary outcome measure of unplanned readmissions seems to be a better primary outcome.

Author response

Given that a safe transition from hospital to home starts with a timely transfer of the patient handover from hospital to the primary care provider (which is now often lacking), we decided to measure the intervention-effect on a timely handover. We agree with the reviewer that our intervention had the potential to address also other outcomes related to the discharge process. Therefore, secondary outcomes were length of hospital stay (LOS) and rates of unplanned readmission within 30 days and we provide the results on these outcome measures as well. Besides, although we understand the reviewer's point of view we believe that it is inappropriate to change our outcome measures as determined in our study protocol.²

2. van Seben R, Geerlings SE, Verhaegh KJ, Hilders CG, Buurman BM. Implementation of a Transfer Intervention Procedure (TIP) to improve handovers from hospital to home: interrupted time series analysis. *BMC health services research* 2016;16:479. doi: 10.1186/s12913-016-1730-x [published Online First: 2016/09/09]

6. The problems with high time measurements before implementation due to increased pre-intervention focus on transferring information quickly should be noted in the limitations.

Author response

We elaborate on this issue in our limitation section now, by stating:

“However, this study design also has limitations. First of all, a positive trend towards sending handovers faster along the pre-intervention period, which was probably due to the attention that was already brought to the discharge process before implementation of the discharge bundle.”

7. In the abstract the sentence “No significant change in levels and slopes was observed in the number of medical and nursing handovers sent within 24h” still will not make sense to people who don't understand what levels and slopes means. Can you just say: ‘No significant change was observed in the number of medical and nursing handovers sent within 24h.’

Author response

We agree with the reviewer that this is still difficult to understand and changed the sentence in our abstract as suggest by the reviewer. Also, throughout the paper we replaced slopes by trends to provide more clarification.

8. There are occasional typos / grammar issues that should be addressed: e.g. ‘Nonetheless, a review of Kripalani et al. showed that medical handover are often not available...’

Author response

We have altered this sentence and now state:

“Nonetheless, a review of Kripalani et al. showed that medical handovers are often not available, lack important information or are not sent in a timely manner.”

Also, we carefully re-read our paper and believe that we solved all typos and grammar issues.

9. You say that ‘Since the study involved a quality improvement intervention with negligible risk of harming patients, individual informed consent was waived for all participating hospitals.’ Can you specify who waived this consent?

Author response

We have added this information and now state:

“Since the study involved a quality improvement intervention with negligible risk of harming patients, individual informed consent was waived for all participating hospitals by the legal department research support of the Amsterdam UMC, location AMC.”

10. A comment about the benefits and drawbacks of measuring the effect of complex interventions through this type of design would be useful with a comparison to other designs which might be more expensive and time consuming – such as cluster trials.

Author response

In our method section we state:

“We evaluated the implementation of the TIP discharge bundle in an interrupted time series (ITS), which is the strongest design when a randomized controlled trial is not feasible”

In our limitation section we come back to our study design now and state:

“An interrupted time series provides a strong quasi-experimental design to evaluate the impact of an intervention aimed at quality improvement. However, this study design also has limitations. First of all, a positive trend towards sending handovers faster along the pre-intervention period, which was probably due to the attention that was already brought to the discharge process before implementation of the discharge bundle. In fact, an important limitation of ITS is that it is more difficult to determine whether the observed effect is a direct effect of the intervention, compared to, for example, clustered trials.”

Reviewer: 4 Julio Díaz

11. From the statistical point of view, the methodology used is correct. However, the following aspects should be explained in more detail. The equations of the adjustment lines must be shown by the method of least squares and indicate the statistical significance of their coefficients, especially the trend.

Author response

We added the equation in our paragraph on statistical analyses:

The segmented regression helped us to estimate the change in the intercept and the slope coefficients between the pre-intervention and post-intervention period using the following model:
$$Y_t = \alpha + \beta_1 \text{time} + \beta_2 \text{intervention} + \beta_3 \text{time after intervention} + \epsilon_t$$

12. The authors on page 10 indicate that they have explored models that have compared with the Akaike Information Criterion. However, these models are not shown in results. This aspect must be clarified and the results shown.

Author response

We thank the reviewer for pointing at this lack of clarification. With regard to our primary outcome measure, we provide the results of the Akaike Information Criterion in the legend of table 2 now:

“a Correction for autocorrelation did not provide a better model compared to the presented model (AIC 74.17 versus 72.88, $p=0.40$), nor did correction for potential confounders (‘polypharmacy’ and

'acute admission') (AIC 74.98 versus 72.88, p=0.39). All models led to results with similar estimates and identical interpretation.

b The results are adjusted for autocorrelation, but not for potential confounders. Correction for autocorrelation (AR1) provided a better model compared to the presented model (AIC 21.52 versus 25.72, p=0.01). Correction for potential confounders ('polypharmacy' and 'acute admission') did not provide a better model compared to the presented model (AIC 29.23 versus 25.72, p=0.78). Correction for autocorrelation (AR1) changed β_1 into a significant result. Correction for potential confounders did not alter the results.

c Correction for autocorrelation did not provide a better model compared to the presented model (AIC 66.05 versus 59.13, p=0.02), nor did correction for potential confounders ('polypharmacy' and 'acute admission') (AIC 59.03 versus 59.13, p=0.13). All models led to results with similar estimates and identical interpretation."

With regard to length of hospital stay and unplanned readmission rates we added the following information in our result section:

"With regard to LOS, the results are adjusted for autocorrelation (AIC 22.64 versus 33.75, p=0.01), but not for potential confounders (AIC 43.08 versus 33.75, p=0.07). With regard to unplanned readmission rates, the results are unadjusted for autocorrelation (AIC 57.18 versus 54.45, p=0.10) and potential confounders (AIC 57.47 versus 54.45, p=0.61)."

VERSION 3 – REVIEW

REVIEWER	Julio Diaz Carlos III Insitute of health
REVIEW RETURNED	15-Feb-2019

GENERAL COMMENTS	Accept
--------